# The Sagittal Integral Morphotype in Male and Female Rowers

**DOI:** 10.3390/ijerph182412930

**Published:** 2021-12-08

**Authors:** Jose Ramón Alvero-Cruz, Fernando Santonja-Medina, Jose Manuel Sanz-Mengibar, Pilar Sainz de Baranda

**Affiliations:** 1Andalucía Tech, Faculty of Medicine, Campus de Teatinos, University of Málaga, 29071 Málaga, Spain; alvero@uma.es; 2Sports and Musculoskeletal System Research Group (RAQUIS), University of Murcia, 30100 Murcia, Spain; jmsmengibar@hotmail.com (J.M.S.-M.); psainzdebaranda@um.es (P.S.d.B.); 3Department of Surgery, Pediatrics, Obstetrics and Gynecology, Faculty of Medicine, University of Murcia, 30100 Murcia, Spain; 4Department of Orthopaedic Surgery and Traumatology, “Virgen de la Arrixaca” University Clinical Hospital, 30120 Murcia, Spain; 5Centre for Neuromuscular Diseases, National Hospital for Neurology and Neurosurgery, University College London Hospitals NHS Foundation Trust, London NW1 2BU, UK; 6Department of Physical Activity and Sport, Faculty of Sport Sciences, University of Murcia, 30100 Murcia, Spain

**Keywords:** anatomy, spine, thoracic spine, low back, lumbar spine, biomechanics, rowing

## Abstract

The goal of this study was to describe the integrated spinal assessment of the sagittal morphotype in rowers to determine whether the intense practice of rowing causes a modification of the sagittal curvatures of the spine, its relationship with the rowing technique, and training background. The second goal was to analyse how the dorsal and lumbar curves behave in the three phases of the rowing gesture, and to determine which phases can be detrimental to the correct development of the spine during growth. We analysed the spine curvatures in the sagittal plane of 29 females and 82 males, which were measured with an inclinometer in standing, slump sitting, maximal trunk flexion and during rowing performance. The average value of thoracic kyphosis in the rowers was 30° (mean, 30 + 8.27°). Thoracic hyperkyphosis was found in only two rowers (1.8%). Lumbar lordosis was within normal range in 84.1% of the males (mean, 27 + 9.57°) and 75.9% of female rowers (mean, 33°). Functional thoracic hyperkyphosis was observed in 57.4% of the males and 17.1% of the females. Functional lumbar hyperkyphosis was observed in 28 of the 69 males (40.5%) and five of 22 females (17.2%). Rowing seems to provide adequate spine alignment in the sagittal plane on standing. The integrated spinal assessment of the sagittal morphotype showed that half or our rowers presented with functional thoracic hyperkyphosis, and 43.2% presented with functional lumbar hyperkyphosis. Spine behaviour during the rowing technique shows that the thoracic kyphosis (98.2%) and lumbar spine (91%) perform within normative ranges and could explain the adequate positioning of the spine in the sagittal plane on standing. Years of rowing training tend to reduce thoracic kyphosis in males.

## 1. Introduction

Research shows that the spine seems to adapt to the biomechanical requirements of different sports [1,2,3], primarily as an engine for the execution of the specific skill, and consequently producing structural changes. Specific training improves performance and may modify the spine curvature according to its intensity [4]. Research has described spinal profiles in artistic [3] and trampoline gymnasts [4], swimmers [5,6,7], climbers [8], tennis players [9], bodybuilders [6,10], skiers [11], dancers [12], and also, riders [13]. The sagittal curvatures of the spine have also been studied in team sports, such as football, rugby [6], hockey players [1], volleyball, basketball, and handball [14]. This adaptative mechanism and its potential consequences [15] have also been studied in rowers, including pain [16,17,18,19], muscle dysfunction due to fatigue [20], and stresses on the intervertebral joints and discs [21]. The wedging of immature vertebral bodies is another potential consequence in high-performance athletes and swimmers [7,22,23] due to repetitive trunk flexion movements [3,24]. The combination of high forces acting on the rower and high training volume put rowers at risk for injury [16]. Kinematics and biomechanical measurements show posterior pelvic tilt and lumbar spine flexion at the catch position [25,26], and fatigue seems to increase this range of motion of the spine [27,28]. Greater bone mineral density [29] has also been used to quantify the mechanical loading produced during rowing.

Research on the sagittal integral morphotype (SIM) in male and female rowers has not been found to exist, neither about the relationship between postural adaptations nor their years of training. At present, thoracic and lumbar kyphosis during rowing technique have not been related to the impact of a standardised clinical spine assessment. The specific impact of this sport on spine alignment requires an assessment that allow comparisons with other disciplines [1,3,4,10,12,13,30,31,32,33]. 

The movement of the lumbar spine in the sagittal plane during the rowing technique has been previously studied [25]. In contrast, the postural adaptation of the spine as a result of rowing training has not been described. Lower back dysfunction due to fatigue [20,27] and stress on the intervertebral joints and discs [21] described in rowers may have an impact on the basal positioning of the spine in the sagittal plane. Sagittal spinal curvatures play an important role in health, and therefore, both clinical assessment and research in this topic seems to be relevant, as in other sports [1,3,4,5,6,7,10,13,30,31,32,33]. 

The “sagittal integral morphotype”, according to Santonja [34], allows a more comprehensive assessment of the thoracic and lumbar curvatures. The curvatures are assessed according to vertebral disposition by three different tests [34,35,36]. This noninvasive assessment of the sagittal curvatures with the inclinometer in standing position, slump sitting, and maximal trunk flexion provides good reproducibility, reliability and correlation with radiographic measurements [37,38]. Pastor [7] compared the diagnosis of thoracic hyperkyphosis, made with an inclinometer, to radiographic findings in young swimmers. A sensitivity of 81.6% and specificity of 100% was found, with an ICC of 0.86 in male swimmers and 0.84 in females. 

The integrated spinal assessment of the sagittal morphotype has proven to be useful in the assessment of the normal status of the spine of different sportsmen [1,3,4,10,12,13,30,31,32,33,35], as well as scholars [35,36,39,40,41] and adolescents [42]. The advantage of this standardised assessment is the possible comparison of the spinal adaptations among different sport techniques. As previously described, many other sport techniques with no similarities with these tests showed compensatory strategies of the spine [1,3,4,5,13] to cope with its biomechanical requirements. The sagittal curves of the spine tend to increase with time, especially in adolescence [7,35,36,39,40,41,42]. 

The goals of this study were to describe the integrated spinal assessment of the sagittal morphotype in male and female rowers, as well as its relationship with each sport technique, years of training and other disciplines. Our hypothesis was that there is a specific adaptation of the spine to the biomechanical requirements of rowing, and this spinal adaptation depends on the years of rowing training. In addition, there exist different adaptations of the integrated spinal assessment of the sagittal morphotype between male and female rowers.

## 2. Materials and Methods

### 2.1. Study Design and Approvals

A cross-sectional analysis was designed to confirm or rule out our hypothesis, describing and correlating the spinal measurements with the years of training of male and female rowers. The journal’s ethical standards were met [43], and this study was approved by the Ethics and Scientific Committee of the University of Murcia (Spain) [ID: 1702/2017]. 

### 2.2. Participants

Participants were recruited from four different clubs associated with rowing federation in the southern region of Spain. Rowers, parents, and coaches were informed of the study procedures before the assessments were performed and all study participants provided written informed consents. Inclusion criteria were: belonging to a male or female rowers’ club with membership in a rowing federation, participating in regional or national competitions, and having trained 18 h per week or more during the last six months [3]. Exclusion criteria were the presence of scoliosis (>20° Cobb angle) or any spine deformity on the sagittal plane that required orthopedic treatment [3]. The final sample included 111 rowers; 29 females and 82 males. The mean age (±standard deviation) was 17.43 ± 3.25 years (men: minimum 14, maximum 35, average 17.2; women: minimum 14, maximum 26, average 17.8) and the mean years of training was 4.98 ± 3.77 (men: minimum 0.5, maximum 20, average 4.72; women: minimum 0.5, maximum 15, 5.73 average). 

### 2.3. Measurements

Integrated spinal assessments of the sagittal morphotype were quantified in all participants [1,3] while wearing undergarments and barefoot. Thoracic and lumbar spine curvatures in the sagittal plane were measured in standing position, slump sitting (SS), and maximal trunk flexion (MTF), according to the assessment protocol defined by Santonja [34]. Measurements were performed by an orthopaedic consultant with 15 years of expertise, and athletes were assessed and data from every rower were obtained during the same session. Quantification of the sagittal curvatures of the spine was carried out with a Unilevel inclinometer (ISOMED, Inc., Portland, OR, USA). Upon standing, the inclinometer was placed at T1 to begin with, and then slid down to the end of the thoracic kyphosis, where the greatest value was observed. After resetting the inclinometer, the lumbar curve was quantified using the same procedure at this vertebral level. In order to quantify the thoracic and lumbar curves in SS and MTF, the inclinometer was placed at T1–T2, T12–L1 and L5–S1 [34,35,36]. Measurements were repeated twice to guarantee reliability (Figure 1). 

The reliability of the researcher who carried out the measurements of the SIM was reflected as ICC: kyphosis in standing = 0.98; lordosis in standing = 0.94; kyphosis in sitting = 0.96; lumbar in sitting = 0.97; kyphosis in flexion = 0.87 and lumbar flexion = 0.97. 

Normal values are considered when the measurements are within normative ranges in standing position, SS and MTF tests [34,35] and are summarised in Table 1. If the thoracic kyphosis was greater than 45°, the measurements in self-correction of the kyphosis were also quantified. “Postural hyperkyphosis” was considered when the hyperkyphosis was reduced to ≤30° during the self-correction tests [34,44]. 

The rowing stroke may be divided into drive and recovery phases. The drive phase includes the ‘catch’ when the oar enters the water, and the lumbar spine is fully flexed at this point of main work [20]. At the end of the drive phase, the oar is withdrawn, switching the lumbar spine to a relatively extended position. The ‘finish’ phase is a mid-way position during the drive and extension phases. Thoracic and lumbar spine measurements were also taken during the sport technique performance on the ergometer and these measurements were obtained during training against the usual resistance used with the ergometer. The measurements were at: (a) ’catch’: maximal flexion of hip and trunk with the seat placed forward; (b) ‘finish’: participants sat upright on the seat, placed at full knee extension and elbows flexed towards the chest; (c) ‘extension’ position: maximal extension of the spine at the end of the stroke [20,27]. The inclinometer was placed and zeroed at T1–T2, before sliding it down to T12–L1 to quantify the thoracic curve. The inclinometer was zeroed again at T12–L1 and slid down to L5–S1 to quantify the lumbar curve. This procedure was carried out in the three positions in order to quantify both curves (Figure 2) [1,3,5,10,12,30,35,44].

Age, gender, and year of training of each rower were also noted in order to correlate these factors to the sagittal plane spine measurements.

### 2.4. Statistical Analysis

All statistical analyses were carried out using MedCalc Statistical Software version 19.0.3 (MedCalc Software bvba, Ostend, Belgium). Initially, an exploratory analysis of the data was carried out, in which central tendency (median and mean), dispersion (95% confidence interval [CI] and standard deviation). Measures were calculated after performance of Shapiro–Wilk’s test to verify normality. The percentages of rowers with spine curvatures outside the normative values were also calculated. An independent samples t-test or Mann-Whitney U test examined differences between genders. The correlations among the variables were examined using the Spearman’s rank correlation coefficients (rho). The magnitude of the correlations was evaluated as trivial (r < 0.10), small (0.10 ≤ r < 0.30), moderate (0.30 ≤ r < 0.50), large (0.50 ≤ r < 0.70), very large (0.70 ≤ r < 0.90), and perfect (r ≥ 0.90) [16]. The acceptable type I error was set at *p* < 0.05 [45].

## 3. Results

The demographic features of the participants, significant differences between male and female rowers, and the average descriptive values of the sagittal curves can be found in Table 2. The independent t-test or Mann-Whitney U test showed no differences in age (*p* = 0.35) or years of training (*p* = 0.49) between the genders. The female participants showed greater lumbar lordotic angles than the males in standing position (*p* = 0.0008), but lower angles of lumbar kyphosis in SS (*p* = 0.028), and MTF (*p* = 0.0026) than the males. Thoracic kyphosis values were similar in standing position between the genders, but lower values in women were observed in SS (*p* = 0.003), and MTF (*p* = 0.045). 

The sport technique analysis showed that the values of thoracic kyphosis during the three phases of rowing (95% CI at ‘catch’, ‘finish’ and ‘extension’) were within the normal range (Table 2) in trunk flexion (≤65°), and were reduced in females during ‘catch’ (*p* = 0.0026) and ‘finish’ (*p* = 0.008). Thoracic extension values are similar in the male and female participants. Only one rower reached 74° during the ‘catch’ phase and 66° during the ‘extension’ phase. Another rower reached 68° during the ‘extension’ phase, but the rest of the participants showed thoracic kyphosis curves within the normal values. 

The 95% CI of the lumbar curve angles were within normal ranges during trunk flexion (10–30°) and were significantly lower in women than in men during the three phases of rowing (*p* = 0.0001). Ten rowers exceeded the normative 30° during the ‘catch’ phase (9%), and a lordotic curve was observed in 25 rowers during the ‘extension’ phase. Table 3 summarises the number of cases according to the classical diagnosis in the standing position of the thoracic and lumbar spines, the sagittal morphotype of the spine integrating the three tests (in the last column of the table), and the percentage of rowers whose measurements were outside the normative values. In total, 91.5% of the males and 86.2% of the females possessed thoracic kyphosis in standing position within the normative values (20–45°). Thoracic hyperkyphosis (>45°) was observed in only one male (50°) and one female rower (55°), but both showed a curve reduction to <30° during the self-correction test [34,44] (postural hyperkyphosis). Of the male participants, 84.1%, and 75.9% of the females showed lumbar lordosis within the normative ranges. Only two males and six female rowers had hyperlordosis (>40°) and all of them presented with postural hyperlordosis (lumbar curve in flexion with kyphosis of 10–30°) [34,35]. 

Regarding the integrative sagittal morphotype, 57.4% of the male rowers had “functional thoracic hyperkyphosis” (normal range kyphosis in standing position with adopted hyperkyphotic attitude during SS and/or MTF; Figure 3), and this finding was three times less frequent in females (17.1%). Twenty-eight of 69 male rowers (40.5%) and five of 22 female rowers (17.2%) with normal lordosis in standing were diagnosed with functional lumbar hyperkyphosis (lumbar kyphotic posture was quantified >30° in the MTF and/or >20° SS tests) (Figure 4).

Correlation values between the studied variables can be found in Table 4 and Figure 5. A weak correlation between thoracic kyphosis and years of training has been observed in the male rowers (rho = −0.258; *p* = 0.02), but it was not significant in the female rowers. Lumbar lordosis in standing position in the male rowers tended to reduce with age (rho = −0.302, *p* < 0.01). Thoracic kyphosis during the three phases of rowing performance tended to reduce with age in the males (‘catch’: rho = −0.419, *p* = 0.0002; ’finish’: rho = −0.229, *p* = 0.04; ‘extension’: rho = −0.337, *p* = 0.0031); and in the ‘finish’ (rho = −0.385; *p* = 0.04) and ‘extension’ (rho = −0.561; *p* = 0.0019) phases in the female rowers.

## 4. Discussion

This research is the first work describing the effect of rowing training on thoracic kyphosis and lumbar lordosis in standing, as well as the effect of rowing training on the SIM of the spine in this sport. Our main relevant findings showed how the kyphosis and lordosis adapt to the rowing technique training. A high percentage of functional thoracic hyperkyphosis was observed (57.4% in the male rowers; three times less in women [17.1%]). Functional lumbar hyperkyphosis was observed in 34.1% of the male rowers, while only half of the female rowers showed this lumbar spine adaptation (17.2%). The female gender seemed to play a protective role for the dynamic adaptation of the spine. Also, our research compared the behaviour of the thoracic and lumbar segments during the three phases of rowing, and our results showed that the thoracic and lumbar spine performed within normal ranges. This could be useful to understand the relationship between back pain and spine adaptations, as well as the need for preventive programmes in this sport.

Low frequencies of thoracic hyperkyphosis in male (1.2%) and female (3.4%) rowers were observed in our study and all cases of thoracic hyperkyphosis were flexible when assessed with self-correction tests [34,44]. This adequate positioning of the thoracic spine in the standing position suggests that rowing is a protective sport against thoracic hyperkyphosis in men and in women. In contrast, 51.8% of hyperkyphosis has been reported in high-level swimmers younger than 15 years old [5,7]. A low frequency of lumbar hyperlordosis was also observed in the male participants (3.7%), indicating that rowing may be beneficial in men with lumbar hyperlordosis. 

### 4.1. Thoracic Kyphosis 

The average value of thoracic kyphosis in standing position was lower in rowers (30°) than in other sports (Table 5). This value was lower than those of artistic [3] and trampoline gymnasts [4], climbers [8], swimmers [5], weightlifters [10], tennis players [31], and hockey players [1]. These values are even more favourable than those of scholars [35,39,40,41] adolescents [42], and non-athletic adults (controls) [12,46]. Thoracic kyphosis of rowers in standing position showed similar values to rhythmic [46] and aesthetic group gymnasts [47]. Only dancers [12,48] showed lower values of thoracic kyphosis in standing, probably due to the posture work and body scheme required in this discipline. No significant differences between genders in this measurement was uncommon. 

### 4.2. Lumbar Spine

The average values of lumbar lordosis in standing were reduced in rowers (males, 27°; females, 33.1°) (Table 2), in comparison to trampoline (40.3°) [4] and beginner rhythmic gymnasts (40.3°) [46], ice-hockey players (38.1°) [22], and swimmers (36.3°, and even 50.9° when measured on radiographs) [7]. Our average values were similar to previous studies in hockey [1], aesthetics gymnasts [47], tennis players [31], kayakers [49], weightlifters [10], swimmers [5] and non-athletic adults [12]. It is important to note that the lumbar hyperlordosis was six times more frequent in females than in males (20.7% vs 3.7%) (Table 5). In contrast, lower values of lumbar hyperlordosis in hockey players (1.4%) [1], show jumping riders (5%) [13], and scholars (9%) [41] have been observed. All cases of lumbar hyperlordosis found in rowers were non-structural.

### 4.3. Sagittal Integral Morphotype

The SIM [34] included the quantification of the curves in at least three different positions (standing, SS and MTF), and implies the definition of new diagnostic concepts. 

### 4.4. Thoracic Morphotype

It is important to note that 46.8% of our rowers (57.5% males, 27.6% females) had normal thoracic spines in standing, but increased position during the flexion and/or sitting tests. This was described as functional thoracic kyphosis by Bado [50]. These percentages were higher than in hockey players (18.9%) [1] and aesthetic gymnasts (25.4%) [47], but lower in artistic gymnasts (men, 65.2%; women, 75%) [3]. The percentages of functional kyphosis in scholars and adolescents without intense sport training were lower, as observed in previous research: 29.7% [42], 36.8% [35], 38.5% [51], and 43.2% [39]. 

The diagnosis of functional thoracic hyperkyphosis is clinically relevant during pubertal growth spurs because curves could become structural, according to Bado [50]. In our rowers, the functional thoracic hyperkyphosis due only to hyperkyphosis in SS represented 84.6% of total functional thoracic hyperkyphosis in rowers, suggesting bad postural hygiene in sitting. In contrast, the functional thoracic hyperkyphosis due to hyperkyphosis in MTF (15.4%) could denote an adaptation to the sport performance. The wedging of immature vertebral bodies is another potential consequence in high-performance rowers, as observed in other young athletes [7,22,23], due to repetitive trunk flexion movements [24]. 

If the sagittal spinal assessment would have been carried out in standing position only, the majority of the rowers would have shown thoracic and lumbar curves within the normative values. This supports the integrated spinal assessment of the sagittal morphotype as a more specific screening tool.

**Table 5 ijerph-18-12930-t005:** Summary of research order by year of publication, including average values of thoracic kyphosis in standing, sitting and maximal flexion of the trunk during the sit and reach test.

		Thoracic Spine			Thoracic Morphotype				Lumbar Spine			Lumbar Morphotype				Demographics	
Group		SP	SSP	MTF	↓	Normal	↑	FTH	SP	SSP	MTF	↓	Normal	↑	FLK	Age, Mean or Range (Years)	n
Rowers	Male	30.2°	47.8°	63.7°	7.5%	91.3%	1.2%	57.5%	27°	20.1°	29.3°	12.5%	83.8%	3.7%	46.3%	17.2 (14–35)	82
	Female	30.6°	39.5°	59.3°	10.3%	86.2%	3.4%	27.6%	33.1°	14°	24.3°	3.4%	75.9%	20.7%	20.7%	17.8 (14–26)	29
Scholars [36]	Both genders				2.2%	70.4%	27.4%	36.8%				1.9%	89.1%	9%	82.3%	8–12	731
In-line hockey [1]	Both genders	38.5°	45°	53.7	1.4%	60.8%	37.8%	18.9%	28.7°	28.7°	31.5°	9.5%	89.2%	1.4%	66.1%	8–15	74
Dressage riders [13]	Both genders	39.2°	34.9°	50.7°	0	61.5%	38.5%	23.10%	40.4°	10°	27.4°	0	46.10%	53.9%	38.50%	9–17	13
Show jumping riders [13]	Both genders	43.8°	44.4°	54.2°	0	50%	50%	40%	43.2°	15.4°	27°	0	50%	50%	40%	9–17	10
Artistic gymnasts [3]	Male	39.6°	26°	62.9°	0%	73.9%	26%	65.2%	27.7°	15.5°	26°	4.3%	78.2%	17.3%	13%	8–30	24
	Female	31.8°	49.3°	61.4°	8.3%	87.5%	4.16%	75%	30.5°	15.7°	27.7°	0%	83.3%	16.6%	29%		24
Scholars [40]	Both genders	35.7°	41.9°	53.9°		71.3%	28.7%	-	32.9°	24.4°	33.4°		73.6%	26.4%	-	8–13	688
Skiers [11]	Both genders	41.2°							33.4°							16–19	51
Aesthetic group gymnastics [47]	Female	29.3°	47.9°	69.1°	22.3%	67%	9.6%	25.4%	32.9°	15.9°	26.4°	6.4%	77.7%	16%	-	10–18	94
Scholars [41]	Male	36.8°	43.7°	55.4°	2.3%	70.2%	27.4%	-	30.9°	26.4°	33.1°	1.9%	89.1%	9%	-	10–18	741
	Female	35.4°	41.8°	54.9°					33.2°	23°	33.5°					10–18	
Ballet dancers [48]		18.5°	6.3°	42.6°	48.6%	51.3%	0%	-	24.7°	1.7°	34.5°	23.7%	75%	1.3%	-	13.2	76
Tennis [31]	Male	43.8°			0%	37.5%	62.5%		27.5°			4.2%	83.35	12.5%	-	13–18	40
	Female	36.1°							32.6°								
Scholars [51]	Male	35.5°	43.1°	64.8°	0%	76.5%	23.5%	38.5%	33.9°	9.2°	19.5°	0%	88.2%	11.8%	20.5–23.9%	11–12	39
	Female	37.5°	49°	68.8°					32.4°	8.7°	16.8°					11–12	46
Teenagers [42]	Male	37.6/47°	43/55.1°	66/80.7°	0%	44.5%	54.5%	29.7%	29/35.7°	7.3/12°	16.6/23°	1.2%	90.5%	8.3%	26.2%	13–18	119
	Female	35/42.5°	37.2/43°	64/73.3°	2.6%	68.6%	29%		34/40.3°	5.8/10°	16.6–18°	3.5%	65.7%	30.8%		13–18	103
Trampoline gymnasts [4]	Male	46.9°	51.3°	62.8°					32°	21°	30.3°					14.9	34
	Female	43°	49.2°	53°					40.3°	14°	25.2°						35
Scholars [52]	Both genders	49.4°			3.5%	24.1%	72.4%	-	49.3°			17.2%	65.5%	17.2%		6–14	58
Weightlifting [10]		40.5°	42.7°	61.6°	0%	72.8%	27.2%	-	31.9°	15.4°	25.4°	0%		18.1%	47.5%	22.8	22
Kayakers [49]	Both genders	42.5°		72.2°					28.6°		35.8°					14–17	30
Dancers [12]	Ballet	28.3°	33.1°	49.7°	18.2%	85.8%	0%	-	35.1°	8.3°	19.8°	0%	84.8%	15.2%	24.3%	17–28 (22.7)	33
	Spanish	22.8°	30.9°	49.4°	48%	52%	0%	-	33.8°	8.3°	19.4°	0%	93.9%	6.1%	12.2%	16–29 (22.1)	33
	Control	37.5	39.7°	71.9°	0%	69.7%	30.3%	-	40.3°	5.5°	15.7°	0%	58.8%	41.2%	9.3%	17–29 (22.7)	33
Rhythmic gymnasts [46]	Beginner	33.4°	37.6°	56.7°	3.7%	82.5%	13.8%	-	40.3°	16.2°	25.1°	1.2%	57.5%	41.3%	-	6–18	81
	Squad	28.3°	38.5°	50.4°	14.6%	80.5%	4.9%	-	35.8°	16.8°	26.3°	3.7%	62.2%	34.1%	-		82
	Control	33.5°	39.5°	59.5°	55	70.9%	24.1%	-	35.3°	13.8°	22.9°	11.4%	63.3%	25.3%	-		79
Swimmers [5]	Male	40.4°		78.4°	1.2%	47%	51.8%	-	31.2°		24.6°	2.3%	82.3%	15.4%	-	9–15	345
	Female	39.5°		73.4°					36.3°		21.6°						
Swimmers [7] (+radiograph)	Male	53.3°			0%	18%	82%	-	43.5°			0%	42%	58%	-	9–15	99
	Female	48.6°			0%	38.8%	61.2%	-	50.9°			0%	18.45	81.6%	-		
Scholars + intervention programme [39]	Intervention	34.1°	46°	60.4°	5.5%	77.8%	16.7%	43.2	29.1°	16.5°	24°	5.5%	94.5%	0%	59.7%	10–11	18
	Control Both genders	35.3–36°	42°	64°	6.1%	66.7%	27.2%		24.8–40°	15–16.5°	28°	7.4%	82.7%	9.95		10–11	81
Scholars [53]	Both genders	42.3°	48.1°	56.6°	5	65.9%	34.1%	-	34.8°	17.2°	28.1°	2.4%	87.8%	9.8%	-		
Adults [54]	Both genders	46.7°		67.4°	0%	24.4%	75.6%	-	32.9°		22.6°	2.4%	81.9%	15.7%	-	19–22	126
Weightlifting [55]	Both genders	46.3°			0%	42.5%	57.5%	-	32.3°			3.8%	83.9%	12.3%	-	18–24	772

Note: Distribution according to the classical thoracic and lumbar morphotypes in standing position and according to the integrated spinal assessment of the sagittal integral morphotype [34,35]. Abbreviations: SP: standing position; SSP: slump sitting position; MTF: maximum trunk flexion; FTH: functional thoracic hyperkyphosis; FLK: functional lumbar hyperkyphosis; C: control; ↑: hyper (kyphosis or lordosis); ↓: hypo (kyphosis or lordosis).

### 4.5. Lumbar Morphotype

The most frequent lumbar integrative sagittal morphotype observed in sportsmen is functional lumbar hyperkyphosis, which involves normal lordotic angles in standing, but excessive kyphotic angles in sitting and/or flexion tests, as described by Santonja et al. [34,35,56]. The behaviour of the lumbar segment in rowers also seems to differ according to gender (Table 2 and Table 5). Functional lumbar hyperkyphosis was observed in 43.2% of our rowers (twice more in men than women). A similar percentage has been observed in weightlifters (47.5%) [10], dressage riders (38.46%), and show jumping riders (40%) [13]. A higher frequency has only been found in hockey players (66.1%) [1]. Lower percentages were present in scholars (20.5–23.9%) [51], artistic gymnasts (12% in men, and 29% in women) [3], and 26% of adolescents [42]. The lowest percentages have been observed in flamenco dancers (flamenco, 12%; ballet, 24.3%) [12]. 

### 4.6. Curve Adaptation According to Sport Technique, Age, and Years of Training

#### 4.6.1. Age and Years of Training

Thoracic kyphosis tends to reduce with years of training in male rowers (rho = −0.258), but not in female rowers. This positive effect could explain why the majority of the male rowers’ thoracic kyphosis measurements (91.3%) fell within the normal range. This is an unusual finding because thoracic kyphosis tends to increase with age [57], and should make us think about the potential benefits of this sport. Lumbar lordosis was not correlated to years of training and tended to reduce with age, only in male rowers (rho = −0.302). 

#### 4.6.2. Sport Technique

Reduced thoracic kyphosis during the three rowing phases in males and during two phases in females (‘finish’ and ‘extension’) was observed. Technique performance seems to perfect with age, leading to less kyphotic positioning; thus, explaining our adaptative results. 

Lumbar lordosis increased only in females during the ‘extension’ phase (rho = 0.503) (X = −5.55 ± 11.1; 95% CI = −8.0–0.0; *p* < 0.0001). Again, this finding supports the benefits of this sport on the spine position in the sagittal plane. Wilson et al. [27] have also shown higher maximal flexion values between L2–L4 during rowing stroke (mean maximum flexion angle, 54.38 ± 9.48° (range, 35.88–75.98°). Differences in spine positioning during rowing between males and females have been observed in our sample in regards to lumbar spine performance, but no differences in age or years of training have been observed. 

The SIM studies the baseline postural adaptation as a result of years of a specific training, not spinal movement during the sport performance. This methodology allows comparison between different sport disciplines, as well as plotting to normative [1,3] and developmental data [58].

The lumbar spine was kyphotic during the three phases of the rowing technique in the male rowers, and in the ‘catch’ and ‘finish’ phases in the females, but all average values were within the normative value range for flexion tests. The average convexity values were greater during ‘catch’, and progressively reduced during the ‘finish’ and ‘extension’ phases for both genders. The male rowers showed significantly more lumbar flexion during the whole performance, and females even showed some degrees of lumbar lordosis at the end of the stroke. 

Our large sample size and no significant differences in terms of age and years of training between the genders allowed us to observe the specific distribution of the sagittal morphotype of the spine in rowers, which has been previously described in other sports (Table 5). Future studies understanding the development of this integrative spinal assessment in regards to years of training, as well as its relationship with back pain in rowers, are recommended. Lumbar curvatures in the slump sitting and trunk forward bending positions, together with height, have been recently found as predicting factors of sciatica history in female classical ballet dancers [32]. Recurrent lower back pain has been related to the lumbar curve during trunk flexion, as well as reduced hamstring extensibility [33,59]. Hamstring length and demands are factors to consider in future analyses of rowers.

Nugent et al. [60] found that rowers without lower back pain, or considered “healthy”, have different kinematics (including flatter low back spinal position at the ‘finish’ phase) than those with lower back pain (larger lumbar kyphosis). McGregor et al. [61] found that rowers with lower back pain had significantly less range of motion at the L5/S1 level (in the ‘catch’ position: 7.5 ± 1.3° in normal; 4.8 ± 1.2° in previous history of lower back pain groups; and 2.8 ± 5.5° in current lower back pain group). Recently, Cejudo et al. [59] found a relationship between lower back pain and static functional lumbar hyperkyphosis, and structured hyperlordosis in male and female team sports players.

Functional adaptations of the spinal sagittal curves seem to be the response to biomechanical stresses [62] and does not always depend on the years of training, but appear to be specific to each sport technique [1,3,8]. Future studies may also describe the effect of preventive intervention for the functional adaptations described. 

The limitations of our study include the lack of radiographic assessment of the three positions, to establish the correlation with inclinometer quantification. Secondly, it has not been possible to measure the sagittal curves dynamically during the rowing performance, as there are no reliable measurement systems that would allow to quantify the curves of the back subjected to the workloads. On the other hand, the reliability of the inclinometer has already been studied against radiographs [7,37,38,63], and ethical limitations were found to irradiate healthy adults for non-clinical reasons. 

The clinical relevancy of our research was to present that rowers are less hyperkyphotic than the general population and other sportsmen (Table 5). This was also observed in the lumbar spine, indicating that rowing may have a lumbar kyphosis-reductive effect in females, while this finding has been found to frequently increase in scholars and teenagers [35]. Regardless of this, spine alignment remained within normative ranges of lumbar kyphosis during the ‘extension’ and ‘finish’ phases of the rowing technique. A total of 9% showed lumbar kyphosis larger than 30° that has been related to lower back pain [60].

Future research will require the determination of the relationship of back pain to the integrated spinal assessment of the SIM in rowers. Also, establishing a specific threshold for lumbar kyphosis during the ‘catch’ and ‘extension’ phases will guide the sport technique to prevent back pain due to functional lumbar hyperkyphosis. 

## 5. Conclusions

Rowing seems to provide appropriate spine positioning in the sagittal plane in the standing position, since 90% of high competition rowers showed kyphosis and lordosis values within the normal range. The SIM indicated that half of the male rowers had thoracic functional hyperkyphosis, and almost half presented with functional lumbar hyperkyphosis. In contrast, lumbar hyperlordosis is the rowing adaptation more often observed in female rowers (20.7%), while functional thoracic hyperkyphosis and functional lumbar hyperkyphosis are less frequently observed in female rowers. 

Spine behaviour during the rowing technique shows that thoracic kyphosis (98.2%) and the lumbar spine (91%) perform within normative ranges, which could explain the adequate positioning of the spine in the sagittal plane upon standing. Years of rowing training tend to reduce thoracic kyphosis in males. 

## Figures and Tables

**Figure 1 ijerph-18-12930-f001:**
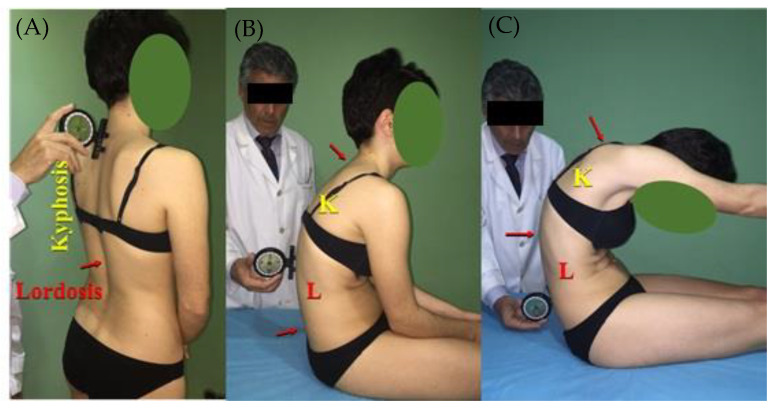
Sagittal integral morphotype of the thoracic and lumbar spine (**A**) Standing assessment: the inclinometer is placed at the beginning of the thoracic kyphosis before zeroing, and it is then slid down to the greatest curve angle is observed. Once the inclinometer is zeroed, the lumbar curve will be quantified with the same procedure. (**B**) Slump sitting quantification: cranial arrow points to the initial placement; the inclinometer was zeroed at T1–T2, before sliding it down to T12–L1 to quantify the thoracic curve. The inclinometer was zeroed again at T12–L1 and slid down to L5–S1 to quantify the lumbar curve. (**C**) Assessment in maximal trunk flexion: both arrows show the thoracic kyphosis limits. K = thoracic kyphosis; L= lumbar curve.

**Figure 2 ijerph-18-12930-f002:**
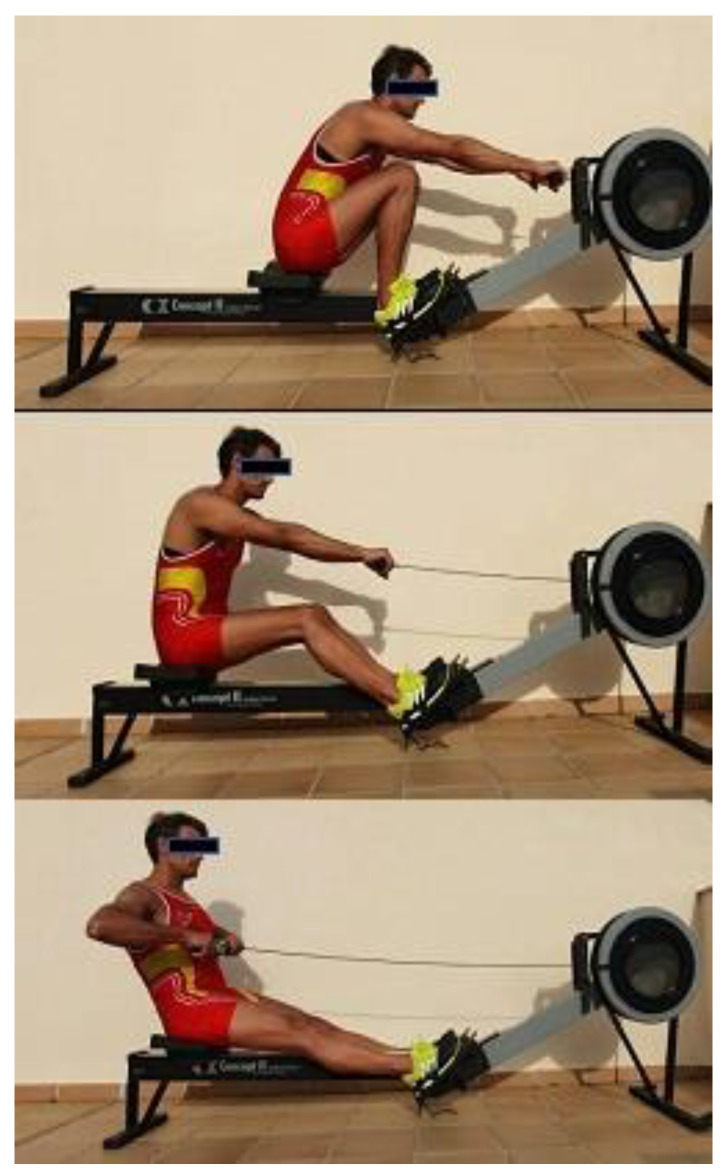
Rowing technique phases. Thoracic and lumbar sagittal curves were measured during the three phases of rowing: catch (**top**), finish (**middle**) and extension (**bottom**). Thoracic kyphosis was quantified placing the inclinometer at T1–T12 and lumbar spine at T12–L5.

**Figure 3 ijerph-18-12930-f003:**
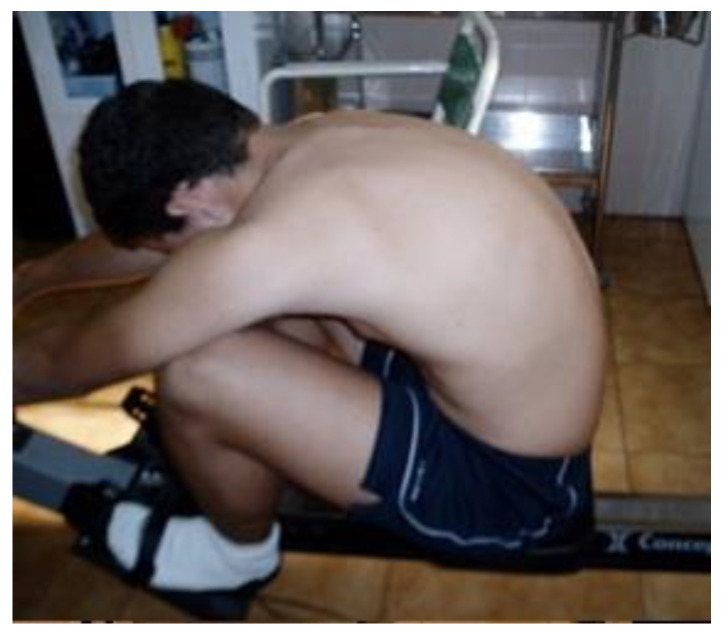
Functional thoracic hyperkyphosis. Normal thoracic kyphosis (30°) in standing was observed in the rower displayed (normal range 20–45°). In maximal trunk flexion, 80° of thoracic kyphosis was quantified, resulting in a functional thoracic hyperkyphosis.

**Figure 4 ijerph-18-12930-f004:**
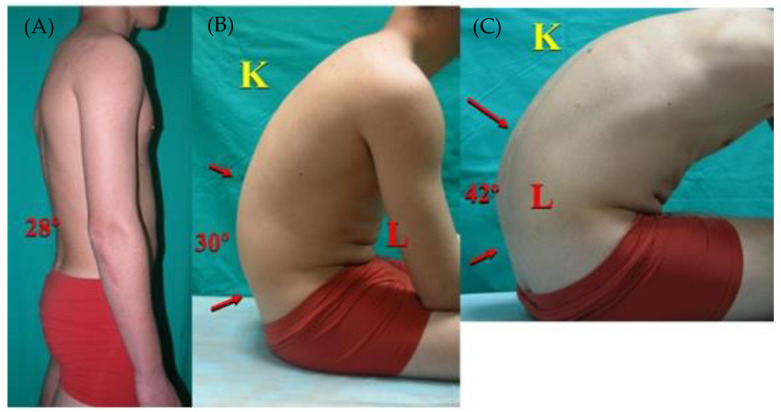
Sportsman with functional lumbar hyperkyphosis (**A**) lumbar lordosis is quantified with the normal range in standing (28°); however, (**B**) lumbar kyphosis is increased in slump sitting (30°), (**C**) as well as in maximal trunk flexion (42°). K = thoracic kyphosis, L = lumbar curve.

**Figure 5 ijerph-18-12930-f005:**
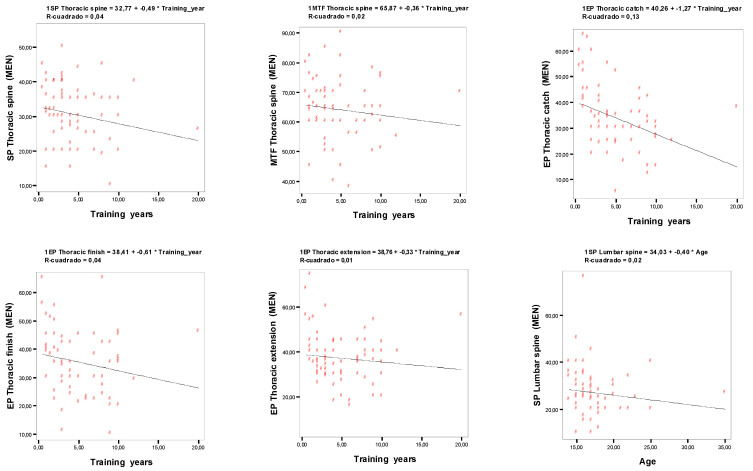
Correlations significative between the sagittal spine curvature values and years of training, age and training technique.

**Table 1 ijerph-18-12930-t001:** Classification of the sagittal thoracic and spine curvatures according to the normative data [34,35].

	Sagittal Morphotype	Thoracic Curvature	Lumbar Curvature
Standing	Normal range	20–45°	(−)20–40°
	Hyperkyphosis/hyperlordosis	>45°	>(−)40°
Maximal trunk flexion in sit and reach test	Normal range kyphosis	40–65°	10–30°
	Thoracic hyperkyphosis	>65°	
	Functional lumbar hyperkyphosis		>0°
Slump sitting	Normal range kyphosis	20–45°	±0–20°
	Thoracic hyperkyphosis/	>5°	>0°
	Functional lumbar hyperkyphosis		

**Table 2 ijerph-18-12930-t002:** Descriptive values: demographic data of the participants and sagittal spine curvatures (significant differences according to gender).

Variables	Men (n = 82)			Women (n = 29)			*p*-Value
	Mean ± SD	Median	95% CI	Mean ± SD	Median	95% CI	
**Demographics**							
Age (years)	17.28 ± 3.23	16	16.0–17.0	17.86 ± 3.34	16.5	16–18.0	0.35
Training (years)	4.72 ± 3.42	4	3.0–5.0	5.73 ± 4.64	4	3–6.6	0.49
**Standing position**							
Thoracic (°)	30.21 ± 8.27	30	29.3–33	30.62 ± 9.03	32	26–35.0	0.82
Lumbar (°)	27.01 ± 9.57	26	25.0–28.0	33.14 ± 9.13	34	30–37.2	0.0008
**Slump sitting**							
Thoracic (°)	47.83 ± 50	10.58	45.6–50.7	39.52 ± 9.48	39	35–45.2	0.003
Lumbar (°)	20.13 ± 9.04	20	16.9–23.0	14.03 ± 11.5	12	7.3–15.0	0.028
**Maximal trunk flexion**							
Thoracic (°)	63.7 ± 9.76	65	61.6–65.0	59.38 ± 10.2	59	55–65.0	0.045
Lumbar (°)	29.35 ± 10.6	26	25.0–30.0	24.31 ± 10.7	22	18–26.0	0.026
**Ergometre position**							
Thoracic catch (°)	34 ± 12.5	31	30.0–35.0	28.31 ± 13.5	25	21–30.2	0.026
Thoracic finish (°)	37.05 ± 11.3	35.5	34.0–40.0	30.07 ± 12.2	26	24.7–35.2	0.0086
Thoracic extension (°)	35.28 ± 11.2	35	30–38.02	36.55 ± 13.3	35	26–42.0	0.86
Lumbar catch (°)	22.24 ± 8.52	22	20.0–25.0	15.45 ± 5.93	16	11.7–20	<0.0001
Lumbar finish (°)	17.67 ± 6.72	18	16.0–20.0	11.79 ± 6.66	12	8.0–15.0	0.0001
Lumbar extension (°)	7.64 ± 8.77	7	5.0–10.0	−5.55 ± 11.1	-5	−8.0–0.0	<0.0001

Abbreviations: SD: standard deviation; CI: confidence interval.

**Table 3 ijerph-18-12930-t003:** Distribution according to the classical thoracic and lumbar morphotypes in standing position and according to the integrated spinal assessment of the sagittal integral morphotype [34,35].

					Male		Female	
					n = 82	%	n = 29	%
CATEGORY Thoracic Morphotype	SUBCATEGORY	Standing	Slump Sitting	Maximal Trunk Flexion	Thoracic Morphotype			
Hypokyphosis or hypokyphotic attitude	Standing	Hypokyphosis (<20°)	Normal (20–45°)	Normal (40–65°)	6 (2 > 45° SS)	7.3	3	10.3
Hypomobile kyphosis		Normal (20–45°)	Normal (20–45°)	Hypokyphosis (<40°)	0		0	
Normal kyphosis		Normal (20–45°)	Normal (20–45°)	Normal (40–65°)	28	34.1	20	68.9
Hyperkyphosis	Total	Hyperkyphosis (>45°)	Hyperkyphosis (>45°)	Hyperkyphosis (>65°)	1	1.2	0	
	Standing	Hyperkyphosis (>45°)	Normal (20–40°)	Normal (40–65°)	0		0	
	Static	Hyperkyphosis (>45°)	Hyperkyphosis (>45°)	Normal (40–65°)	0		1	3.4
	Dynamic	Hyperkyphosis (>45°)	Normal (20–40°)	Hyperkyphosis (>65°)	0		0	
Functional thoracic hyperkyphosis	Static	Normal (20–45°)	Hyperkyphosis (>45°)	Normal (40–65°)	40	48.8	4	13.7
	Dynamic	Normal (20–45°)	Normal (20–40°)	Hyperkyphosis (> 65°)	3	3.7	1	3.4
	Total	Normal (20–45°)	Hyperkyphosis (>45°)	Hyperkyphosis (>65°)	4	4.9	0	
**CATEGORY** **Lumbar morphotype**	**SUBCATEGORY**	**Standing**	**Slump sitting**	**Maximal trunk flexion**	**Lumbar morphotype**			
Hypolordosis	Lumbar hypomobility	Hypolordotic attitude (<20°)	Normal (0 ± 20°)	Normal (10–30°)	1	1.2	0	
Normal lordosis		Normal (20–40°)	Normal (0 ± 20°)	Normal (10–30°)	41	50	17	58.6
Functional lumbar hyperkyphosis	Static	Normal (20–40°)	Hyperkyphosis (>20°)	Normal (10–30°)	9	11	0	
	Dynamic	Normal (20–40°)	Normal (0 ± 20°)	Hyperkyphosis (>30°)	3	3.6	0	
	Total	Normal (20–40°)	Hyperkyphosis (>20°)	Hyperkyphosis (10–30°)	16	19.5	5	17.2
Hyperlordosis	Postural or attitude	>40°	Normal (0 ± 20°)	Normal (10–30°)	2	2.4	6	20.7
	Structural	> 40°	Normal (0 ± 20°) or lordotic (<−20°)	Hypokyphosis (<10°)	0		0	
Lumbar hypermobility		Hyperlordosis (>40°)	Normal (0±20°) or hyperkyphosis (>20°)	Normal (10–30°) or hyperkyphosis (> 30°)	1	1.2	0	
Lumbar kyphosis		Hypolordosis or kyphosis (< 20°)	Hyperkyphosis (>20°)	Hyperkyphosis (>30°)	9	11	1	3.4
Structural lumbar kyphosis		Lumbar kyphosis	Hyperkyphosis (>20°)	Hyperkyphosis (>30°)	0		0	

Note: comparisons according to the classical thoracic and lumbar spine morphotypes in standing position only and the integrative assessment of the sagittal morphotype of the spine (standing, slump sitting and maximal trunk flexion) in male and female rowers.

**Table 4 ijerph-18-12930-t004:** Correlations between the sagittal spine curvature values and years of training, age and training technique.

	Men		Women	
Variable	Training Years	Age	Training Years	Age
**Standing position**				
Thoracic spine	−0.258 *	0.021	0.018	0.033
Lumbar spine	−0.204	−0.302 **	−0.161	−0.045
**Slump sitting**				
Thoracic spine	−0.069	−0.038	−0.031	0.086
Lumbar spine	0.097	−0.034	0.304	0.15
**Maximal trunk flexion**				
Thoracic spine	−0.249 *	0.135	−0.203	−0.293
Lumbar spine	0.115	−0.031	0.041	−0.266
**Ergometre position**				
Thoracic catch	−0.419 **	−0.216	−0.261	−0.173
Thoracic finish	−0.229 *	0.053	−0.385 *	−0.24
Thoracic extension	−0.337 **	0.083	−0.561 **	−0.309
Lumbar catch	-0.026	-0.168	0.104	0
Lumbar finish	0.056	0.008	−0.237	−0.118
Lumbar extension	0.212	−0.009	0.503 **	0.047

Note: * *p* < 0.05, ** *p* < 0.01.

## Data Availability

The data used to support the findings of this study are available from the corresponding author upon request.

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
