# Peer review of "The Sagittal Integral Morphotype in Male and Female Rowers"

_ijerph, 2021, doi:10.3390/ijerph182412930_

Round 1

Reviewer 1 Report

thank you for your effort with the updated version of your manuscript

Author Response

COMENTARIOS REVISOR 1

Comments and Suggestions for Authors

thank you for your effort with the updated version of your manuscript

Thank you very much for your positive comments

Reviewer 2 Report

L85 - reviewer's question from original review not addressed.

L100 - same as above

L134-150 I'm still not sure how the measurements are representative of 'sporting actions' when several thousand newtons of force are not going through the body. Am I right in that this is what they're supposed to represent? Can the authors outline why there is the assumption that there would be no difference between and unloaded static position and a loaded dynamic position?

L249 why are these graphs unavailable?

Results (including ones presented in abstract). Were measurements made to two decimal places. If only measured to a whole degree, please report with zero decimal places to represent the sensitivity of the measures.

Author Response

Dear Editor and reviewers of the manuscript entitled The Sagittal Integral Morphotype in male and female rowers”.

We appreciate your observations and the time devoted to the constructive criticism and feedback of our manuscript. We improved different aspects of the discussion section and conclusions to draw attention to the relevant results and compare previous results in the literature, as you propose.

Thank you for your comments and the possibility to address the potential reader’s questions.  

English language has now been edited (please find attached the certificate), and we have highlighted the changes in red in the manuscript.

COMENTARIOS REVISOR 1

Comments and Suggestions for Authors

thank you for your effort with the updated version of your manuscript

Thank you very much for your positive comments

COMENTARIOS REVISOR 2

L85 - No rationale provided as why the authors would expect to see a training-age related relationship with spinal curvature provided in intro. What is the relationship expected to be with age – Linear, curvelinear?

Studies carried out by our research group (7,35,36,39,40,41,42,52) indicate that hyperkyphosis and hyperlordosis tend to increase throughout growth. Based on these data, our hypothesis was that there would be a similar tendency in the rowers, that is, to increase the degree of thoracic kyphosis with age. Likewise, we thought that rowing should be a sport that could increase kyphosis, due to its technique, so we thought that the longer hours and the older the rowers (and consequently, more years of training), the more degrees of increased kyphosis should be observed.  

The following text has been added to the introduction section: “The sagittal curves of the spine tend to increase with growth especially in adolescence (7,35,36,39-41,42).”  line 88 to 89.

L100 - How does chronological age (teenage through to mature adult) affect spinal curvature?

As described above, several studies point towards an increasing tendency of hyperkyphosis and hyperlordosis with growth (7,35,36,39-41,42). This is also our clinical experience as health professionals, with a more remarkable tendency during puberty.

L134-150 I'm still not sure how the measurements are representative of 'sporting actions' when several thousand newtons of force are not going through the body. Am I right in that this is what they're supposed to represent? Can the authors outline why there is the assumption that there would be no difference between and unloaded static position and a loaded dynamic position?

We agree that our measurements were taking “statically” because, as far as we know, there is no other validated measurement method of the sagittal curves other than with devices need that required to be attached on the back (at the beginning and at the end of the thoracic and lumbar curves). Systems based on kinematic analysis through videos or photographs are less precise (lack of precision selecting the beginning and the end of each curve, as well as limited capture of the spinal processes due overlapping with the erector spinae muscles in a lateral view.

Regarding the effect of the load on the maximum kyphosis obtained in the three phases of rowing, all our measurements of the kyphosis and the lumbar curve during rowing technique were obtained against the usual resistance used in their training with the ergometer.

The following text has been added to the Method and Measurements section: “These measurements were obtained during training against the usual resistance used with the ergometer. The measurements were at:” (line 155 to 157).

In our opinion, the maximum degree of kyphosis in trunk flexion does not vary with load. The maximum kyphosis adopted in trunk flexion is the result of the maximum amplitude of the spine that is limited by resistance to compression forces at the anterior part of the vertebrae, and pulling forces to which the posterior vertebral structures are subjected. This resistance is constant (unalterable). If this resistance gives way, a wedging fracture of the vertebral body and / or a rupture of the posterior vertebral ligaments would occur. In other words, in order for the degree of kyphosis (dorsal and / or lumbar) to increase when carrying weight or when performing the rowing gesture, at least one of these structures would have to collapse, which would inevitably cause symptoms that would make it impossible to practice rowing. We verified this when measuring dorsal and lumbar kyphosis in trunk flexion in bodybuilders, while performing exercises lifting weight with trunk flexion, as in the “deadlift” exercise when measuring curves with and without weight (López Miñarro, Segura [10]), not observing significant changes in the degree of maximum dorsal kyphosis in trunk flexion.

The following text has been added to the discussion section: “Secondly, it has not been possible to measure the sagittal curves dynamically during the rowing performance, as there are no reliable measurement systems that would allow to quantify the curves of the back subjected to the workloads”.  (lines 419 to 422).

L249 why are these graphs unavailable? Would prefer to see correlation scatter graphs for some of the relationships rather than a table of correlations.

Following your suggestion, we have now added a Figure 5 with the more significant correlations (L253 to 255). We are sorry that these graphs were not available in the first instance; our statistician is terminally ill, but we had now an external consultancy.

Results (including ones presented in abstract). Were measurements made to two decimal places. If only measured to a whole degree, please report with zero decimal places to represent the sensitivity of the measures.

We agree that inclinometers measure up to full degrees only, and they cannot provide decimals figures. Our decimals figures were a statistical approximation to show mean values with the greatest precision. Following your suggestion, our mean values have been now modified

This manuscript is a resubmission of an earlier submission. The following is a list of the peer review reports and author responses from that submission.

Round 1

Reviewer 1 Report

I appreciate the authors' contribution to the preparation of this paper, both in terms of the literature review and the precision of the description of the material and methods and the discussion. 
I have only minor comments on the manuscript; see below:

Abstract

1) line 31: please explain the word: kyperkyphosis?

Materials and methods section:

1) Was it randomized selection ?

2) Study participants were man and woman aged 17.43 ± 3.25 years. Authors write, that "...study participants provided written informed consent". I would suggest to add clarification that such consent was provided either by the participants (I guess below 18 y.o.) or by the guardians of the participants (18+)

3) line 101: it is written: "... orthopedic treatment. 3 The final sample included 111 rowers..." What does "3" mean?

4) line 119: "Measurements were repeated twice to guarantee their reliability". Which measurement was taken into acount?

Author Response

Dear Editor and reviewers of the manuscript entitled The Sagittal Integral Morphotype in male and female rowers”.

We appreciate your observations and the time devoted to the constructive criticism and feedback of our manuscript. We improved different aspects of the discussion section and conclusions to draw attention to the relevant results and compare previous results in the literature, as you propose.

Thank you for your patient and understanding of the time needed to reply to your comments. Three other articles from our research team and same topic were awaiting final approval, and they have now been accepted in different journals and included in this article. These articles were relevant to back up your observations in topics like the relation to the sagittal curves and pain.

English language has now been edited (please find attached the certificate), and we have highlighted the changes in red in the manuscript.

Abstract

1) line 31: please explain the word: kyperkyphosis? This has now been clarified since the new classification of the integral sagittal morphotype was published (34,35), and more specifically in table 1.  

Materials and methods section:

1) Was it randomized selection ? The selection was not randomized,  all the participants were included according to inclusion and exclusion criteria

2) Study participants were man and woman aged 17.43 ± 3.25 years. Authors write, that "...study participants provided written informed consent". I would suggest to add clarification that such consent was provided either by the participants (I guess below 18 y.o.) or by the guardians of the participants (18+). This has now been ammended in the text

3) line 101: it is written: "... orthopedic treatment. 3 The final sample included 111 rowers..." What does "3" mean? This mistake now been amended and clarified in the text

4) line 119: "Measurements were repeated twice to guarantee their reliability". Which measurement was taken into acount?

Both measurements should agree, otherwise a new measurement will be repeated. The reliability of the assessor was good enough to find twice the same quantification. The inclinometer is an instrument for assessment that is widely used in clinical practice. It quantifies the sagittal curvatures of the spine (the thoracic kyphosis and the lumbar lordosis), allowing comparison and differentiation between individuals with normal curves and those with pathological curves (Hyper or hypo). This significantly reduces the number of radiographic studies required for both diagnosis and follow-up. Furthermore, it is widely used in the bibliography (in PubMed, 190 articles are obtained when searching for “inclinometer and kyphosis”, and 216 for “inclinometer and lordosis”).

Another utility of the inclinometer is the measurement of trunk mobility, which is why it is used for patients with low back pain (in PubMed; inclinometer and low back pain = 830). These data show that many clinicians and researchers are interested in research using this instrument.

Regarding its validity, we conducted a study with 99 growing elite swimmers, which represented a study sample with the 345 best swimmers in Spain. These 99 randomly selected swimmers underwent a standing lateral spinal X-ray, after measuring their thoracic kyphosis and lumbar lordosis in the same position. When the X-ray obtained a Kyphosis Cobb > 40º, and the thoracic Kyphosis measured with the inclinometer was also > 40º, the sensitivity was 77.5% and the specificity was 100%. If we raised the angular value of Khyphosis in the Rx > 45º, and that of kyphosis with an inclinometer was also > 40º, the sensitivity increased to 81.6% and the specificity remained at 100% (Pastor, 2000). The statistical correlation between the clinical test with an inclinometer and the radiographic value for the thoracic kyphotic curve in standing was 0.86 for swimmers and 0.84 for female swimmers.

The following text has been add to the manuscript: “Pastor (7), compared the diagnosis of thoracic hyperkyphosis made with an inclinometer to the radiographic findings in young swimmers.  A sensitivity 81.6% and Specificity of 100% was found, with and ICC of 0.86 in male swimmers and 0.84 in females” (Line 79-82).

Reviewer 2 Report

L21 change ‘sport’ technique to ‘rowing’ technique

L22 change ‘quantified’ to ‘analysed’

L26 change ‘ (average 27o)’ to the mean and standard deviation e.g. ‘ (27±3o)’

Abstract lacks conclusion. Rather than just reporting the results there needs to be some information of what the results mean and their context.

L38 ‘primary’ should be ‘primarily’

L37-46 Need more background information. How exactly does the spine ‘adapt’ to these sports? Please could the authors describe some of these adaptations or results from these studies on various athletes.

L54 Please define what the sagittal integral morphotype actually means

L56-61 Repetitive information.

L75 change ‘proofed’ to ‘proved’

L76 change ‘sportsman’ to ‘sportsmen’ or athletes

L62-68 provides the main rationale for the study. However it is too brief and needs to be elaborated upon. Other sections of the introduction can be written more concisely to allow this extra information. Currently the rationale for the study is relatively weak with no major focus on the value of the outcomes of the tests.

L83-89 The hypothesis and study design are incompatible because there is no non-rowing control group provided. Therefore, it can’t satisfy the first part of the hypothesis which is that rowing causes specific spinal adaptations because it’s a cross section of rowers only.

L85 No rationale provided as why the authors would expect to see a training-age related relationship with spinal curvature provided in intro. What is the relationship expected to be with age – Linear, curvelinear?

L99-105 Are these the only exclusion criteria? What about illness/ injury, neurological disorders etc??

L100 How does chronological age (teenage through to mature adult) affect spinal curvature?

L134-150 It appears the ‘rowing technique’ was assessed from static positions. How are these positions valid or representative for ‘rowing technique’ when the structures aren’t going through the high forces and load that actually occurs during the rowing stroke which the authors note in the introduction? Should these not have been done dynamically and through some kinematic analysis?

L249 Would prefer to see correlation scatter graphs for some of the relationships rather than a table of correlations.

Table 5 – I would expect to see this type of table in a systematic review not an original investigation. This information needs to be condensed and described succinctly in the text and table removed.

I think all reference within this paper to measures made during the ‘rowing technique’ should be removed. The authors need to demonstrate that these measures are valid and reliable in terms of a rower performing a dynamic, high load rowing stroke.

Author Response

Dear Editor and reviewers of the manuscript entitled The Sagittal Integral Morphotype in male and female rowers”.

We appreciate your observations and the time devoted to the constructive criticism and feedback of our manuscript. We improved different aspects of the discussion section and conclusions to draw attention to the relevant results and compare previous results in the literature, as you propose.

Thank you for your patient and understanding of the time needed to reply to your comments. Three other articles from our research team and same topic were awaiting final approval, and they have now been accepted in different journals and included in this article. These articles were relevant to back up your observations in topics like the relation to the sagittal curves and pain.

English language has now been edited (please find attached the certificate), and we have highlighted the changes in red in the manuscript.

Comments and Suggestions for Authors

L21 change ‘sport’ technique to ‘rowing’ technique. This has now been changed in the text, in red.

L22 change ‘quantified’ to ‘analysed’. This has now been changed in the text, in red.

L26 change ‘ (average 27o)’ to the mean and standard deviation e.g. ‘ (27±3o)’ This has now been changed in the text, in red.

Abstract lacks conclusion. Rather than just reporting the results there needs to be some information of what the results mean and their context. Conclusion has now been improved in the abstract, in red.

L38 ‘primary’ should be ‘primarily’. This has now been changed in the text, in red.

L37-46 Need more background information. How exactly does the spine ‘adapt’ to these sports? Please could the authors describe some of these adaptations or results from these studies on various athletes.

There are various sports that are "detrimental" for the correct development of the sagittal curvatures of the spine (Weightlifting [55], In-line Hockey [1], Dressage riders [13], Show jumping riders [13], Artistic Gymnasts [3]. Other activities such as Ballet [48] [12], have shown a beneficial effect on the thoracic kyphosis. In the case of swimming, frequently prescribed in back problems, has unexpectedly shown a high frequency of thoracic kyperkyphosis in young elite swimmers 51.8% [Pastor, 2002].

  1. Sainz de Baranda P, Cejudo A, Moreno-Alcaraz VJ, et al. Sagittal spinal morphotype assessment in 8 to 15 years old InlineHockey players. PeerJ 2020;8.
  2. Sanz-Mengibar JM, Sainz-de-Baranda P, Santonja-Medina F. Training intensity and sagittal curvature of the spine in male and female artistic rowers. J Sport Med Phys Fit 2018; 58(4): 465–471.
  3. Sainz de Baranda P, Santonja Medina F, Rodríguez-Iniesta M. Tiempo de entrenamiento y plano sagital del raquis en gimnastas de trampolín. Rev Int Med Cienc Act Fis Dep 2010; 10 (40): 521-536.
  4. Ginés-Díaz A, Martínez-Romero MT, Cejudo A, et al. Sagittal spinal morphotype assessment in dressage and show jumping riders. J Sport Rehabil 2019; 1–23.
  5. Santonja-Medina F, Collazo M, Martínez-Romero MT, R et al. Classification System of the Sagittal Integral Morphotype in Children from the ISQUIOS Programme (Spain). Int J Environ Res Public Health 2020; 17(7):2467.
  6. Sainz de Baranda P, Cejudo A, Martínez-Romero MT, et al. Sitting posture, Sagittal Spinal Curvatures and Back Pain in 8 to 12-Year-Old Children from the Region of Murcia (Spain): ISQUIOS Programme. Int J Environ Res Public Health 2020; (7):2578.
  7. Martínez-Gallego F, Rodríguez P. Metodología para una gimnasia rítmica saludable. Madrid: Consejo Superior de De-portes; 2006.
  8. Conesa E. Valoración de la columna en el plano sagital y extensibilidad isquiosural en Gimnasia Estética de Grupo. (in Spanish) [PhD Thesis]. Murcia University: Murcia, Spain: 2015.
  9. Vaquero-Cristóbal R, Esparza-Ros F, Gómez-Durán R, et al. Morfología de las curvaturas torácica y lumbar en bipe-destación, sedentación y máxima flexión del tronco con rodillas extendidas en bailarinas. Arch Med Deport 2015. 32: 87–93.
  10. López Miñarro PA. Análisis de ejercicios de acondicionamiento muscular en salas de musculación, incidencia sobre el raquis en el plano sagital. (in Spanish) [PhD Thesis]. Murcia University: Murcia, Spain: 2003.

Please find this comment more developed in the text in red

L54 Please define what the sagittal integral morphotype actually means.

The spine assessment according to the sagittal integral morphotype, includes the quantification of the thoracic and lumbar curves in three different positions: standing, slump sitting and sit a reach test. This facilitates specific functional diagnosis in relation to the sport performance [1,3,4,10,12,13,30-33]. Please find this comment more developed in the text in red

L56-61 Repetitive information. This has now been amended

L75 change ‘proofed’ to ‘proved’. This has now been amended

L76 change ‘sportsman’ to ‘sportsmen’ or athletes. This has now been amended

L62-68 provides the main rationale for the study. However it is too brief and needs to be elaborated upon. Other sections of the introduction can be written more concisely to allow this extra information. Currently the rationale for the study is relatively weak with no major focus on the value of the outcomes of the tests.

The key motivation behind our focus was to know if the practice of rowing (performed regularly and intensely) modifies the development of the sagittal curvatures of the spine.

The significance of our study is to provide new information about the behavior of the sagittal curves of the spine during rowing, but also the implications of intense rowing training on the position of the thoracic and lumbar spine. In reference to the relation between the sagittal integral morphotype and back pain, we have just published 3 articles that can be relevant to back up your comment. 

The following text has now been added to the manuscript:

“Lumbar curvature in slump sitting and trunk forward bending positions, together with height, have been recently found as predictor factors of sciatica history in female classical ballet dancers [56]. Recurrent lower back pain has been related to the lumbar curve during trunk flexion, as well as reduced hamstring extensibility [57,58]. Hamstrings length and demands are factors to consider in future analysis of rowers.” (Lines 386 to 392).

“Nugent et al [59] found that rowers without lower back pain or “healthy”, have different kinematics (including flatter low back spinal position at the finish phase) that those with lower back pain (larger lumbar kyphosis). McGregor et al [60] found that rowers with lower back pain had significantly less range of motion at L5/S1 level (in the catch position 7.5°± 1.3 in normals; 4.8°±1.2 in previous history of lower back pain groups; and 2.8° ± 5.5 in current lower back pain group). Cejudo et al (2021) have found a relation between lower back pain and static functional lumbar hyperkyphosis, and structured hyperlordosis in male and female team sports players [57].” (Lines 393 to 400).

L83-89 The hypothesis and study design are incompatible because there is no non-rowing control group provided. Therefore, it can’t satisfy the first part of the hypothesis which is that rowing causes specific spinal adaptations because it’s a cross section of rowers only.

In relation to the lack of control group, there are different studies with control samples in the same area where our rowers research took part [10,12, 35,39,40,41,42,47,51,52,53, 54]. Some of these previous studies included even similar group ages and were used as control group. Table 5 compares all the studies published to date, including those with the control sample. This control group shows significantly differences in the assessment of the sagittal integral morphotype compared to our male and female high competition rowers.

  1. Segura D. Programa de musculación para desalineaciones sagitales del raquis en adultos jóvenes. (in Spanish) [PhD Thesis]. Murcia University: Murcia, Spain: 2009.
  2. Gómez Lozano S, Vargas-Macías A, Santonja-Medina F, Canteras M. Estudio descriptivo del morfotipo raquídeo sagital en bailarinas de flamenco. Rev Cent Investig Flamenco Telethusa 2013; 6 (7): 19-28.
  3. Santonja-Medina F, Collazo M, Martínez-Romero MT, Rodríguez-Ferrán O, Cejudo A, Andújar P, Sainz de Baranda P. Classification System of the Sagittal Integral Morphotype in Children from the ISQUIOS Programme (Spain). Int J En-viron Res Public Health 2020; 17(7):2467.
  4. Sainz de Baranda P. Programa para la mejora del raquis en el plano sagital y extensibilidad isquiosural en enseñanza primaria. (in Spanish) [PhD Thesis]. Murcia University: Murcia, Spain: 2002.
  5. Santonja Renedo F. Efectos de un programa de educación postural sobre el morfotipo sagital del raquis y la extensibi-lidad isquiosural: estudio multicéntrico en escolares de Educación Primaria. (in Spanish) [PhD Thesis]. Murcia Univer-sity: Murcia, Spain: 2017.
  6. Collazo M. Morfotipos sagitales del raquis en población escolar en la Región de Murcia. (in Spanish) [PhD Thesis]. Murcia University: Murcia, Spain: 2015.
  7. Peña WA. Educación Física y Salud: Programa para la mejora del raquis en el plano sagital y la extensibilidad isquio-sural en Secundaria. (in Spanish) [PhD Thesis]. Murcia University: Murcia, Spain: 2010.
  8. Conesa E. Valoración de la columna en el plano sagital y extensibilidad isquiosural en Gimnasia Estética de Grupo. (in Spanish) [PhD Thesis]. Murcia University: Murcia, Spain: 2015.
  9. Fernández Campos MJ. Efecto de un Programa de Educación Postural en Educación Primaria: tres años de seguimiento. (in Spanish) [PhD Thesis]. Murcia University: Murcia, Spain: 2011.
  10. Andújar P. Prevalencia de las desalineaciones sagitales del raquis en edad escolar en el municipio de Murcia. (in Spa-nish) [PhD Thesis]. Murcia University: Murcia, Spain: 2010.
  11. Martínez-García AC. Efectos de un programa de educación postural sobre el morfotipo sagital del raquis, la extensibi-lidad de la musculatura isquiosural y psoas iliaco y la resistencia muscular abdominal y lumbar en escolares de Educa-ción Secundaria (in Spanish) [PhD Thesis]. Murcia University: Murcia, Spain: 2013.
  12. Ríos de Moya Angeler R. Evaluación a los nueve años del programa de atención al niño estudio de factores antropo-métricos, cardiovasculares y de la columna vertebral. (in Spanish) [PhD Thesis]. Murcia University: Murcia, Spain: 2012.

L85 No rationale provided as why the authors would expect to see a training-age related relationship with spinal curvature provided in intro. What is the relationship expected to be with age – Linear, curvelinear?

L99-105 Are these the only exclusion criteria? What about illness/ injury, neurological disorders etc?? This has now been amended in the text

L100 How does chronological age (teenage through to mature adult) affect spinal curvature?

L134-150 It appears the ‘rowing technique’ was assessed from static positions. How are these positions valid or representative for ‘rowing technique’ when the structures aren’t going through the high forces and load that actually occurs during the rowing stroke which the authors note in the introduction? Should these not have been done dynamically and through some kinematic analysis?

Regarding rowing, thoracic hyperkyphosis could be empirically develop during the “drive” phase, having an impact on the normal spine development. Sport practice can modify the Sagittal Morphotype of the spine, as shown in previous research [1,3,4,13,46,47]. Since the new classification of the integral sagittal morphotype was published (34,35), this statement has been further corroborated.

L249 Would prefer to see correlation scatter graphs for some of the relationships rather than a table of correlations. These graphs are unfortunately unavailable

Table 5 – I would expect to see this type of table in a systematic review not an original investigation. This information needs to be condensed and described succinctly in the text and table removed. We agree, table 5 compares all the studies published to date, including those with the control sample. This control group shows significantly differences in the assessment of the sagittal integral morphotype compared to our male and female high competition rowers. Table 5 was included in order to back up and give reader a quicker access to previous sport measurements.
